# The genes significantly associated with an improved prognosis and long-term survival of glioblastoma

Hong Gyu Yoon[1], Jin Hwan Cheong[2], Je Il Ryu[2], Yu Deok Won[2], Kyueng-Whan Min[3], Myung-Hoon Han[2]*

1 Department of Neurosurgery, Yonsei University College of Medicine, Seoul, Republic of Korea,
2 Department of Neurosurgery, Hanyang University Guri Hospital, Hanyang University College of Medicine, Guri, Gyeonggi-do, Republic of Korea, 3 Department of Pathology Uijeongbu Eulji Medical Center, Eulji University School of Medicine, Uijeongbu, Gyeonggi-do, Republic of Korea

* gksmh80@gmail.com

**Data Availability Statement:** All relevant data are within the paper and its Supporting Information files.

## Abstract

### Background and purpose

Glioblastoma multiforme (GBM) is the most devastating brain tumor with less than 5% of patients surviving 5 years following diagnosis. Many studies have focused on the genetics of GBM with the aim of improving the prognosis of GBM patients. We investigated specific genes whose expressions are significantly related to both the length of the overall survival and the progression-free survival in patients with GBM.

### Methods

We obtained data for 12,042 gene mRNA expressions in 525 GBM tissues from the Cancer Genome Atlas (TCGA) database. Among those genes, we identified independent genes significantly associated with the prognosis of GBM. Receiver operating characteristic (ROC) curve analysis was performed to determine the genes significant for predicting the long-term survival of patients with GBM. Bioinformatics analysis was also performed for the significant genes.

### Results

We identified 33 independent genes whose expressions were significantly associated with the prognosis of 525 patients with GBM. Among them, the expressions of five genes were independently associated with an improved prognosis of GBM, and the expressions of 28 genes were independently related to a poorer prognosis of GBM. The expressions of the *ADAM22*, *ATP5C1*, *RAC3*, *SHANK1*, *AEBP1*, *C1RL*, *CHL1*, *CHST2*, *EFEMP2*, and *PGCP* genes were either positively or negatively related to the long-term survival of GBM patients.

### Conclusions

Using a large-scale and open database, we found genes significantly associated with both the prognosis and long-term survival of patients with GBM. We believe that our findings may contribute to improving the understanding of the mechanisms underlying GBM.

**Funding:** This study was funded by the Basic Science Research Program through the National Research Foundation of Korea (NRF) funded by the Ministry of Science, ICT & Future Planning (NRF-2022R1F1A1063739).

**Competing interests:** The authors have declared that no competing interests exist.

## Introduction

Glioblastoma multiforme (GBM) is the most common and devastating primary brain tumor, which is characterized by infiltrative growth and resistance to treatment and leads to an extremely poor prognosis. Despite aggressive treatment strategies against GBM, including chemotherapy, radiotherapy, immunotherapy, and surgical resection, only a few patients survive 2.5 years, and less than 5% of patients survive 5 years following their diagnosis [1].

Extensive studies have focused on the genetics of GBM to improve the understanding of the underlying mechanisms of GBM and to contribute to an improved prognosis of patients with GBM [2]. We also previously identified a DKK3 gene from the Wnt/β-catenin pathway and 12 genes from 10 oncogenic signaling pathways associated with GBM prognosis using The Cancer Genome Atlas (TCGA) database [3, 4]. It is well known that TCGA is the world's largest publicly accessible genomic database. It includes information on digital pathologic slides, mRNA expression data, clinicopathological information, and DNA methylation and mutation data. However, there has not been a study aiming to identify the genes significantly related to the prognosis of GBM by assessing the direct association between the gene expression levels in GBM tissue and both the lengths of the overall survival (OS) and the progression-free survival (PFS) in patients with GBM, using large gene expression datasets of GBM. In addition, we hypothesize that if genes related to long-term survival in patients with GBM are found, it may help predict the future prognosis or treatment of patients with GBM.

Therefore, this study aimed to investigate specific genes, using the TCGA database, whose expressions are significantly related to both the lengths of OS and PFS in patients with GBM. Next, we aimed to classify the identified genes significantly associated with the prognosis of GBM, according to the Gene Ontology (GO) terms using bioinformatics. Finally, this study aimed to identify which genes, among the identified genes significantly related to the GBM prognosis, were significantly associated with the long-term survival of patients with GBM. A schematic flow chart depicting the steps involved in this research is presented in **Fig 1**.

## Methods

### Study patients

We obtained 1,149 glioma cases, consisting of 619 GBM cases and 530 low-grade glioma cases with mRNA gene expression data from the TCGA database (https://gdc.cancer.gov/about-data/publications/pancanatlas and https://www.cbioportal.org/) [5]. We initially selected 594 GBM cases with virtual histopathological slides and clinical data out of 619 GBM cases. We excluded 594 GBM cases with significantly incomplete mRNA gene expression information and clinical data. Therefore, the 525 GBM cases with complete virtual histopathological slides, mRNA expression data, and clinical information were finally included in the study as described elsewhere [3, 4]. Log 2 (x + 1) transformation normalized all mRNA gene expression values before analysis [6].

Informed consent was not required because the data were obtained from the publicly available TCGA database.

### Study design

In **Fig 1**, the study design is shown as follows: (1) we initially observed a dataset from the TCGA database containing mRNA expression information for 12,042 genes from 525 GBM tissues; (2) then excluded 11,187 genes whose expressions showed no significant association with the lengths of the OS or PFS in the study's patients, according to Pearson correlation analysis (p ≥ 0.01); (3) excluded 819 genes with a low strength of correlation: Genes showing a Pearson coefficient absolute value of less than 0.2, according to a previous study [7]; (4) after

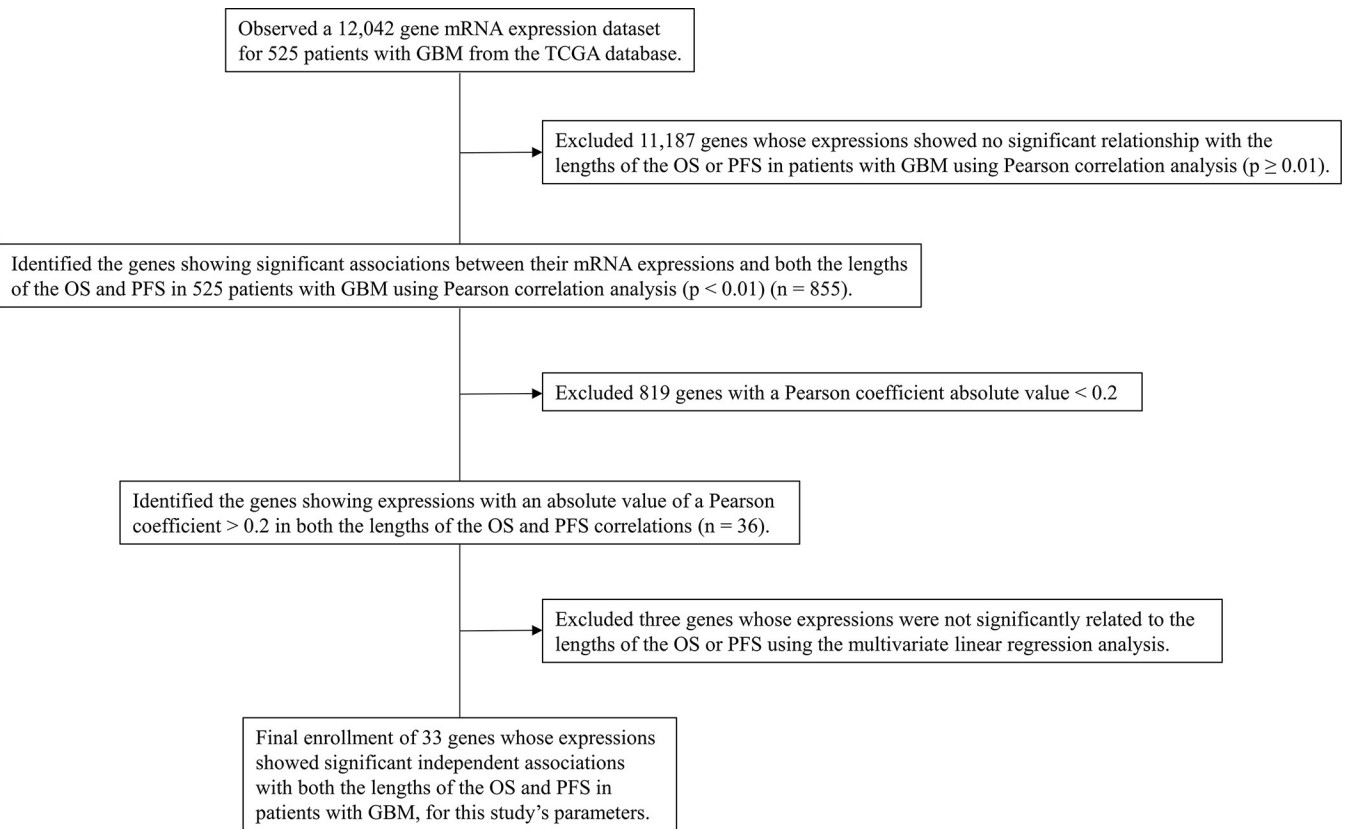

**Fig 1. Schematic diagram detailing the process of selecting the independent genes significantly associated with the prognosis of GBM for our study.**

adjusting for clinical variables, three genes whose expressions were not significantly associated with the lengths of the OS or PFS were further excluded (**Table 1**);

(5) A total of 33 genes whose expressions showed significant independent associations with both the lengths of the OS and PFS in patients with GBM were finally enrolled for the study. We also present the results of the univariate linear regression analysis of the lengths of the OS and PFS according to the 36 significant gene expressions in patients with GBM in the S2 Table. The raw data related to the study design can be found in the S1 Data.

## In silico flow cytometry

As previously reported, we analyzed tumor-infiltrating lymphocytes in GBM tissues using CIBERSORT (https://cibersort.stanford.edu), a versatile computational method for quantifying the immune cell-type fractions. This method relies on a validated leukocyte gene signature matrix containing 547 genes and 22 human immune cell subpopulations [3, 4]. The gene expression profiles of the GBM tissues from the TCGA were entered into CIBERSORT for analysis, and the algorithm was run using the LM22 signature matrix at 100 permutations.

CD8+ T-cells are major drivers of antitumor immunity, and elevated CD8+ T-cell counts in the tumor microenvironment are related to a good prognosis in cancer [8]. In addition, as we have previously described, CD4+ T-cells, CD8+ T-cells, regulatory T-cells (Tregs), B-cells, and antigen-presenting cells are reported to play an important role in the immune microenvironment of GBM [3]. Therefore, we included the following eight representative immune cells for the study to evaluate the relationships between the status of the GBM immune

**Table 1. Multivariable linear regression analysis of the lengths of the OS and PFS according to the 36 significant genes in patients with GBM.**

| Variable | Multivariable linear regression analysis* | | | |
| | Length of OS (months) | | Length of PFS (months) | |
| | β (95% CI) | p-value | β (95% CI) | p-value |
|---|---|---|---|---|
| ADAM22 | 11.99 (4.36 to 19.62) | 0.002 | 10.73 (5.04 to 16.42) | < 0.001 |
| AEBP1 | −1.72 (−2.94 to −0.51) | 0.006 | −1.31 (−2.22 to −0.40) | 0.005 |
| ATP5C1 | 6.37 (2.77 to 9.98) | 0.001 | 4.35 (1.64 to 7.06) | 0.002 |
| C13orf18 | −2.21 (−3.72 to −0.70) | 0.004 | −2.27 (−3.39 to −1.15) | < 0.001 |
| C1RL | −2.70 (−4.25 to −1.15) | 0.001 | −1.95 (−3.11 to −0.79) | 0.001 |
| CBR1 | −2.04 (−3.76 to −0.33) | 0.020 | −2.44 (−3.71 to −1.17) | < 0.001 |
| CCL2 | −1.62 (−2.53 to −0.71) | 0.001 | −1.05 (−1.74 to −0.37) | 0.003 |
| CHI3L1 | −0.84 (−1.56 to −0.13) | 0.022 | −1.04 (−1.57 to −0.51) | < 0.001 |
| CHL1 | −1.56 (−2.52 to −0.61) | 0.001 | −1.58 (−2.29 to −0.88) | < 0.001 |
| CHST2 | −2.17 (−3.71 to −0.62) | 0.006 | −2.15 (−3.30 to −1.00) | < 0.001 |
| CLEC5A | −1.83 (−3.20 to −0.47) | 0.009 | −1.87 (−2.89 to −0.85) | < 0.001 |
| DHRS2 | 7.03 (4.39 to 9.67) | < 0.001 | 4.15 (2.14 to 6.15) | < 0.001 |
| DYNLT3 | −2.78 (−4.64 to −0.92) | 0.004 | −3.66 (−5.02 to −2.30) | < 0.001 |
| EFEMP2 | −3.14 (−4.76 to −1.51) | < 0.001 | −2.59 (−3.80 to −1.38) | < 0.001 |
| EMP3 | −1.52 (−2.72 to −0.33) | 0.013 | −1.62 (−2.51 to −0.74) | < 0.001 |
| F3 | −1.86 (−3.40 to −0.31) | 0.019 | −2.13 (−3.28 to −0.99) | < 0.001 |
| FBXO17 | −2.40 (−4.54 to −0.26) | 0.028 | −2.71 (−4.30 to −1.11) | 0.001 |
| FLJ11286 | −2.41 (−4.35 to −0.46) | 0.015 | −2.63 (−4.07 to −1.18) | < 0.001 |
| **KIAA0495** | **−3.66 (−7.45 to 0.124)** | **0.058** | **−5.06 (−7.86 to −2.25)** | **< 0.001** |
| MSN | −2.97 (−4.85 to −1.09) | 0.002 | −2.84 (−4.24 to −1.44) | < 0.001 |
| NSUN5 | −4.32 (−6.94 to −1.70) | 0.001 | −3.62 (−5.58 to −1.66) | < 0.001 |
| PDPN | −1.31 (−2.34 to −0.29) | 0.012 | −1.32 (−2.08 to −0.55) | 0.001 |
| PGCP | −3.62 (−5.78 to −1.46) | 0.001 | −3.04 (−4.65 to −1.43) | < 0.001 |
| PPCS | −4.26 (−6.54 to −1.98) | < 0.001 | −4.02 (−5.70 to −2.33) | < 0.001 |
| RAC3 | 6.72 (3.26 to 10.17) | < 0.001 | 4.93 (2.33 to 7.52) | < 0.001 |
| **RANBP17** | **4.31 (−1.15 to 9.76)** | **0.121** | **4.27 (0.19 to 8.35)** | **0.040** |
| **RBP1** | **−0.79 (−1.84 to 0.27)** | **0.143** | **−1.38 (−2.16 to −0.60)** | **0.001** |
| SERPING1 | −1.98 (−3.17 to −0.78) | 0.001 | −1.35 (−2.25 to −0.46) | 0.003 |
| SHANK1 | 16.42 (8.33 to 24.50) | < 0.001 | 12.70 (6.64 to 18.75) | < 0.001 |
| SLC25A20 | −3.22 (−5.42 to −1.02) | 0.004 | −2.60 (−4.25 to −0.96) | 0.002 |
| SLC2A10 | −1.70 (−3.19 to −0.22) | 0.025 | −2.02 (−3.12 to −0.91) | < 0.001 |
| STEAP3 | −1.96 (−3.49 to −0.43) | 0.012 | −2.42 (−3.55 to −1.29) | < 0.001 |
| SWAP70 | −3.36 (−5.43 to −1.28) | 0.002 | −3.26 (−4.80 to −1.72) | < 0.001 |
| TIMP1 | −1.75 (−3.19 to −0.32) | 0.017 | −2.17 (−3.23 to −1.11) | < 0.001 |
| TMEM22 | −2.17 (−3.73 to −0.61) | 0.007 | −2.21 (−3.37 to −1.05) | < 0.001 |
| TRIP6 | −2.07 (−3.88 to −0.27) | 0.024 | −2.04 (−3.38 to −0.69) | 0.003 |

OS: overall survival; PFS: progression-free survival; CI: confidence interval.

The rows containing genes showing p ≥ 0.05 in overall survival or progression-free survival of multivariable linear regression analysis are shown in bold.

*adjusted for sex, age, Karnofsky performance scale score, and radiation treatment.

microenvironment and specific gene expressions: CD8+ T-cells, regulatory T-cells, naive CD4 + T-cells, resting and activated memory CD4+ T-cells, memory B-cells, plasma B-cells, and activated dendritic cells [3].

## Bioinformatics analysis

We performed bioinformatics analysis using Cytoscape (version 3.9.1) software (https://cytoscape.org/). We used ClueGo and CluePedia plugins that enabled functional Gene Ontology and pathway network analyses in Cytoscape to interpret the biological roles and interactions of the 33 selected significant genes in GBM [9]. We analyzed the biological function annotated pathways based on 33 significant genes related to the prognosis of GBM. We also activated the cerebral view function in the ClueGO application of the Cytoscape to estimate the approximate location of any significant proteins in the cell. We also conducted pathway-based network analysis using the Search Tool for the Retrieval of Interacting Genes/Proteins (STRING) database version 11.5 (http://www.string-db.org/) to further investigate the inter-relationship between these 33 significant gene expressions. The STRING provides known and predicted protein-protein association data from a large database based on co-expression analysis, signals across genomes, and automatic text-mining of the biomedical literature. All interaction sources, text-mining, experiments, databases, co-expression, neighborhood, gene fusion, and co-occurrence were activated in the STRING setting.

## Statistical analysis

Heatmap analyses of 33 significant gene expressions and immune cell infiltrations in 525 GBM tissues were performed using R software's "pheatmap" package (version 4.1.2).

Pearson correlation coefficients and significance levels were calculated to evaluate the associations between the 33 significant gene expressions and the lengths of the OS and PFS in patients with GBM and the immune cell infiltrations in GBM tissues. We used the "corrplot" package of R software with the clustering technique (R code: corrplot, M, order = "hclust", sig. level = 0.01, method = "square") to visualize the correlations. A scatterplot with a linear regression line was used to visualize the relationship between several significant gene expressions and the lengths of the OS and PFS in patients with GBM. The OS and PFS months were transformed to the natural log scale to normalize the distributions for the analysis. We calculated the OS and PFS rates using Kaplan–Meier analysis based on the gene expression quartiles in patients with GBM.

Receiver operating characteristic (ROC) curve analysis was performed to determine the genes significant for predicting the 2.5-year and 5-year survivals in patients with GBM, defined as showing the shortest distance from the upper left corner (where sensitivity = 1 and specificity = 1).

A p-value < 0.05 was considered statistically significant. All statistical analyses were performed using R software version 4.1.2 and SPSS for Windows version 24.0 (IBM, Chicago, IL).

## Results

### Characteristics of the study patients

A total of 525 patients with GBM from the TCGA database were included in this study. The mean patient age at the diagnosis of GBM was 57.7 years, and 39.0% of patients were female. A total of 435 (82.9%) patients underwent radiation treatment, and further detailed information, including immune cell fractions in GBM tissues, is shown in the S1 Table.

### Identification of significant genes associated with the prognosis of GBM

Through the process shown in **Fig 1**, among the 12,042 observed genes, we identified 33 independent genes whose mRNA expressions were significantly associated with both the lengths of the OS and PFS in patients with GBM. The identified 33 independent and significant genes

are: *ADAM22*, *AEBP1*, *ATP5C1*, *C13orf18*, *C1RL*, *CBR1*, *CCL2*, *CHI3L1*, *CHL1*, *CHST2*, *CLEC5A*, *DHRS2*, *DYNLT3*, *EFEMP2*, *EMP3*, *F3*, *FBXO17*, *FLJ11286*, *MSN*, *NSUN5*, *PDPN*, *PGCP*, *PPCS*, *RAC3*, *SERPING1*, *SHANK1*, *SLC25A20*, *SLC2A10*, *STEAP3*, *SWAP70*, *TIMP1*, *TMEM22*, and *TRIP6*. Among these 33 genes, there were 5 genes (*ADAM22*, *ATP5C1*, *DHRS2*, *RAC3*, and *SHANK1*) whose expressions were positively correlated with the lengths of the OS and PFS. The expressions of the remaining 28 genes exhibited negative correlations with the lengths of the OS and PFS in patients with GBM.

## Expression patterns of the 33 significant genes and immune cells in GBM

The heat map shows different mRNA expression patterns between the 33 significant genes in 525 GBM tissues (**Fig 2A**).

GBM: glioblastoma multiforme; OS: overall survival; PFS: progression-free survival

There were three genes whose mRNA expression levels were noticeably increased in GBM, and those genes were *ATP5C1*, *CHI3L1*, and *TIMP1* (**Fig 2B**). Among the five genes associated with a good prognosis of GBM, the expressions of *DHRS2*, *ADAM22*, *RAC3*, and *SHANK1* were relatively reduced in GBM tissues. The heatmap also showed differences in eight immune cell fractions between the 525 GBM tissues (**Fig 2C**). Heterogenous infiltrations were observed in CD8+ T-cells, resting CD4+ T-cells, naive CD4+ T-cells, and memory B-cells between the 525 GBM tissues. Boxplots show overall fractional differences between eight representative immune cells in the GBM tissues (**Fig 2D**).

## Correlations between the expressions of the 33 genes, the lengths of the OS and PFS, and the immune cells in GBM

We visualized the correlations between the mRNA expressions of the 33 significant genes and the lengths of the OS and PFS in patients with GBM (**Fig 2E**). The expressions of 5 genes (*ADAM22*, *ATP5C1*, *DHRS2*, *RAC3*, and *SHANK1*) showed positive correlations with the lengths of the OS and PFS by providing Pearson coefficients greater than 0.2. The remaining 28 genes showed negative correlations with the lengths of the OS and PFS, providing Pearson coefficients less than –0.2. When we estimated correlations between the expressions of the 33 significant genes and the infiltrations of the eight immune cells from the 525 GBM tissues, there were significant correlations ($p < 0.01$) between the expressions of the 32 genes and the CD8+ T-cell infiltrations, except for the *ATP5C1* gene (an x in the box indicates a p-value $\geq 0.01$) (**Fig 2F**). We also found that the expressions of C13orf18, CHI3L1, CHL1, and CHST2 showed significant correlations with all eight immune cell fractions in GBM.

## Associations between the expressions of the selected genes and the lengths of the OS and PFS in patients with GBM

We observed significant positive linear associations between the expression of ADAM22, ATP5C1, RAC3, and SHANK1 and the lengths of the OS and PFS in patients with GBM (**Fig 3A**).

GBM: glioblastoma multiforme; OS: overall survival; PFS: progression-free survival; ROC: receiver operating characteristic.

Using the Kaplan–Meier survival analysis, the fourth quartiles of ADAM22, ATP5C1, RAC3, and SHANK1 expressions showed significantly greater OS and PFS rates than those in the first, second, and third quartiles, except for the fourth quartile analysis of RAC3 for PFS ($p = 0.1$) (**Fig 3B**). Among the 28 genes associated with poor prognosis of GBM, we observed that the expressions of C13orf18, CHI3L1, CHL1, and CHST2, which were associated with all eight immune cell fractions, showed significant negative linear associations with the lengths of

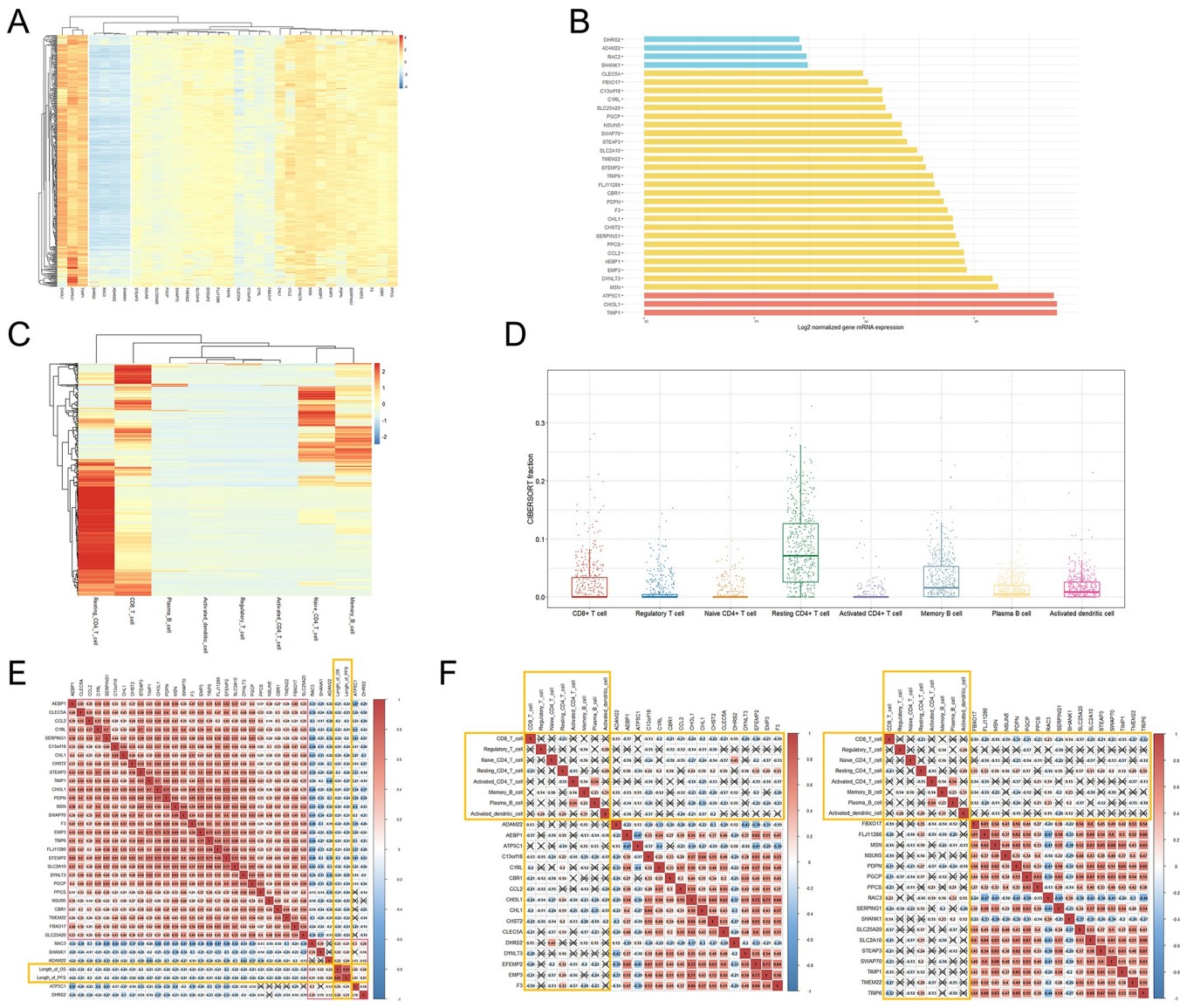

**Fig 2. Gene expression patterns of the 33 independent and significant genes with comparisons of the immune cell fractions in GBM.** The correlations between the 33 significant genes, the OS and PFS lengths, and fractions of representative immune cells in GBM. (A) A hierarchically clustered heatmap showing the expression patterns of the 33 significant genes related to the prognosis of GBM. Gene expression levels were log2 transformed, and a color density indicating levels of log2 fold changes is presented. Red and blue represent up- and downregulated expression, respectively, in GBM; (B) a bar plot indicating average expression levels of the 33 significant genes in GBM tissue; (C) a hierarchically clustered heatmap showing the expression patterns of eight representative immune cells in GBM; (D) boxplots showing the differences in eight representative immune cell fractions in GBM; (E) Pearson correlation coefficients and significance levels were calculated between the expressions of the 33 significant genes and lengths of the OS and PFS in patients with GBM; (F) Pearson correlation coefficients and significance levels were calculated between the expressions of the 33 significant genes and fractions of representative eight immune cells in GBM. The color-coordinated legend indicates the value and sign of the Pearson correlation coefficient. The number in the box indicates the Pearson correlation coefficient. The 'x' in the box indicates a p-value $\geq 0.01$.

the OS and PFS in patients with GBM (**Fig 3C**). The first quartiles of C13orf18, CHI3L1, CHL1, and CHST2 expressions were significantly associated with greater OS and PFS rates compared to other quartile groups (**Fig 3D**). We also analyzed the OS and PFS in patients with GBM according to the quartile groups of the remaining 25 gene expressions, which are not included in the main figures (S1 and S2 Figs). We observed that both OS and PFS were statistically significant in all the remaining genes except for DHRS2 and SWAP70.

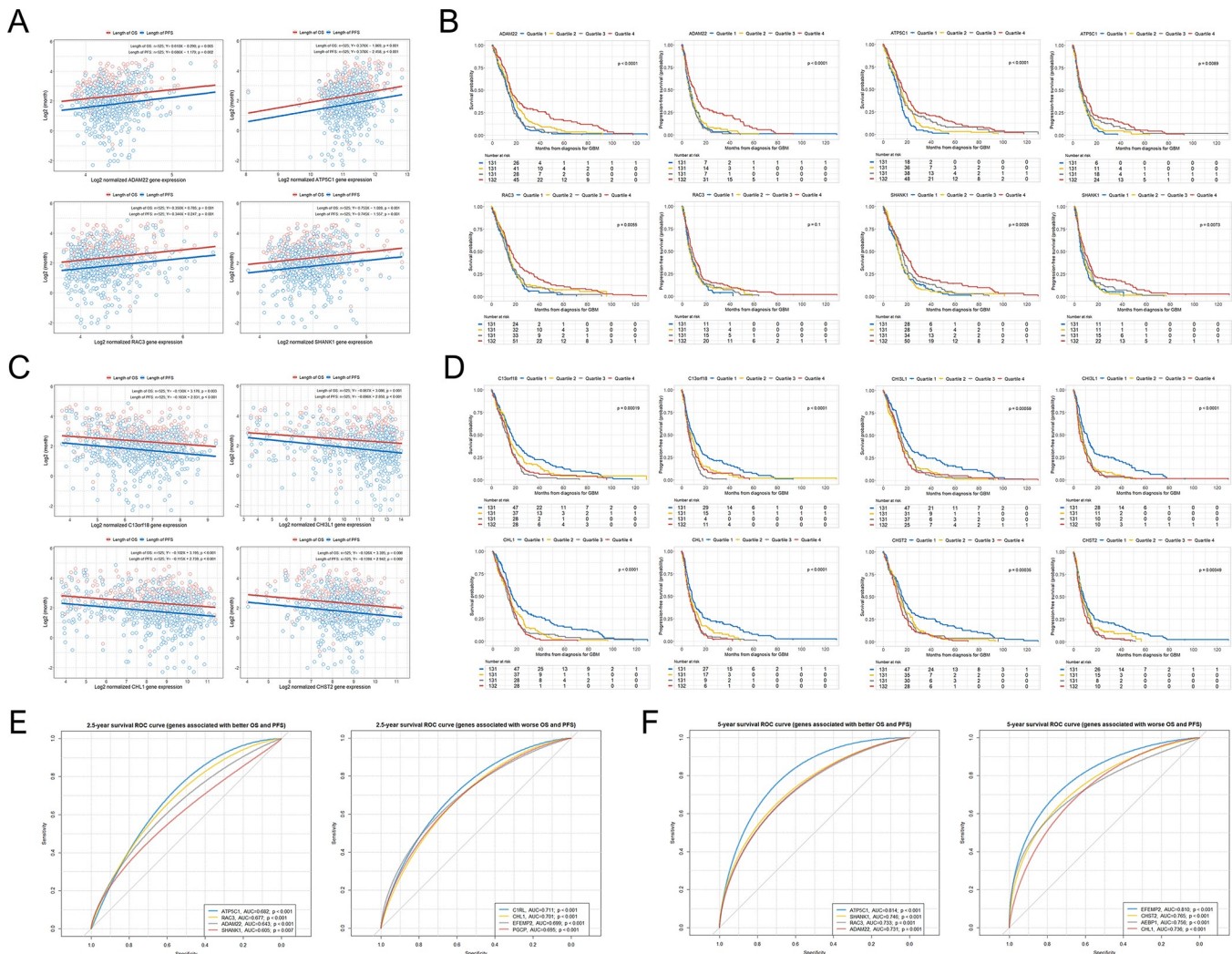

**Fig 3. Scatter plot with linear regression line between several significant gene expressions and log2-transformed lengths of the OS and PFS in patients with GBM.** Kaplan–Meier analysis showing the OS and PFS rates based on several significant gene expression quartiles in patients with GBM. The ROC curves to identify significant genes associated with 2.5-year and 5-year survivals in patients with GBM. (A) Linear regression lines showing the associations between ADAM22, ATP5C1, RAC3, and SHANK1 expressions and the lengths of the OS and PFS in patients with GBM; (B) Kaplan–Meier curves showing the OS and PFS rates according to ADAM22, ATP5C1, RAC3, and SHANK1 expression quartiles in patients with GBM; (C) linear regression lines showing the associations between C13orf18, CHI3L1, CHL1, and CHST2 expressions and the lengths of the OS and PFS in patients with GBM; (D) Kaplan–Meier curves showing the OS and PFS rates according to C13orf18, CHI3L1, CHL1, and CHST2 expression quartiles in patients with GBM; (E) ROC curves showing the significant genes both positively and negatively associated with a 2.5-year survival in patients with GBM; (F) ROC curves showing the significant genes both positively and negatively associated with 5-year survival in patients with GBM.

## Identification of genes whose expressions predict long-term survival of patients with GBM

According to the ROC analysis of our study, when only the top four genes with the highest area under the curve (AUC) were included, higher expressions of ATP5C1 (AUC = 0.682; $p < 0.001$), RAC3 (AUC = 0.677; $p < 0.001$), ADAM22 (AUC = 0.643; $p < 0.001$), and SHANK1 (AUC = 0.605; $p = 0.007$), and lower expressions of C1RL (AUC = 0.711; $p < 0.001$), CHL1 (AUC = 0.701; $p < 0.001$), EFEMP2 (AUC = 0.699; $p < 0.001$), and PGCP (AUC = 0.695; $p < 0.001$) in GBM tissues were associated with the long-term survival (more than 2.5 years) in patients with GBM (**Fig 3E**). When predicting the long-term survival of

more than 5 years in patients with GBM, the identification of higher expressions of ATP5C1 (AUC = 0.814; p < 0.001), SHANK1 (AUC = 0.746; p < 0.001), RAC3 (AUC = 0.733; p = 0.001), and ADAM22 (AUC = 0.731; p = 0.001), alongside lower expressions of EFEMP2 (AUC = 0.810; p < 0.001), CHST2 (AUC = 0.765; p < 0.001), AEBP1 (AUC = 0.756; p < 0.001), and CHL1 (AUC = 0.736; p < 0.001), in GBM tissue, provided significant associations with a long-term survival (more than 5 years) in patients with GBM (**Fig 3F**).

## Functional gene ontology and pathway network analyses

The ClueGO and the CluePedia plugins of Cytoscape were performed to identify the enriched pathways to investigate the functionally grouped networks of the 33 significant proteins in GBM. We found three significant GO terms, which are 'neuromuscular process controlling balance', 'mitochondrial proton-transporting ATP synthase complex, catalytic sector F(1)', 'carbonyl reductase (NADPH) activity' among the five significant proteins (ADAM22, ATP5C1, DHRS2, RAC3, and SHANK1) associated with an improved prognosis of GBM (**Fig 4A and 4B**).

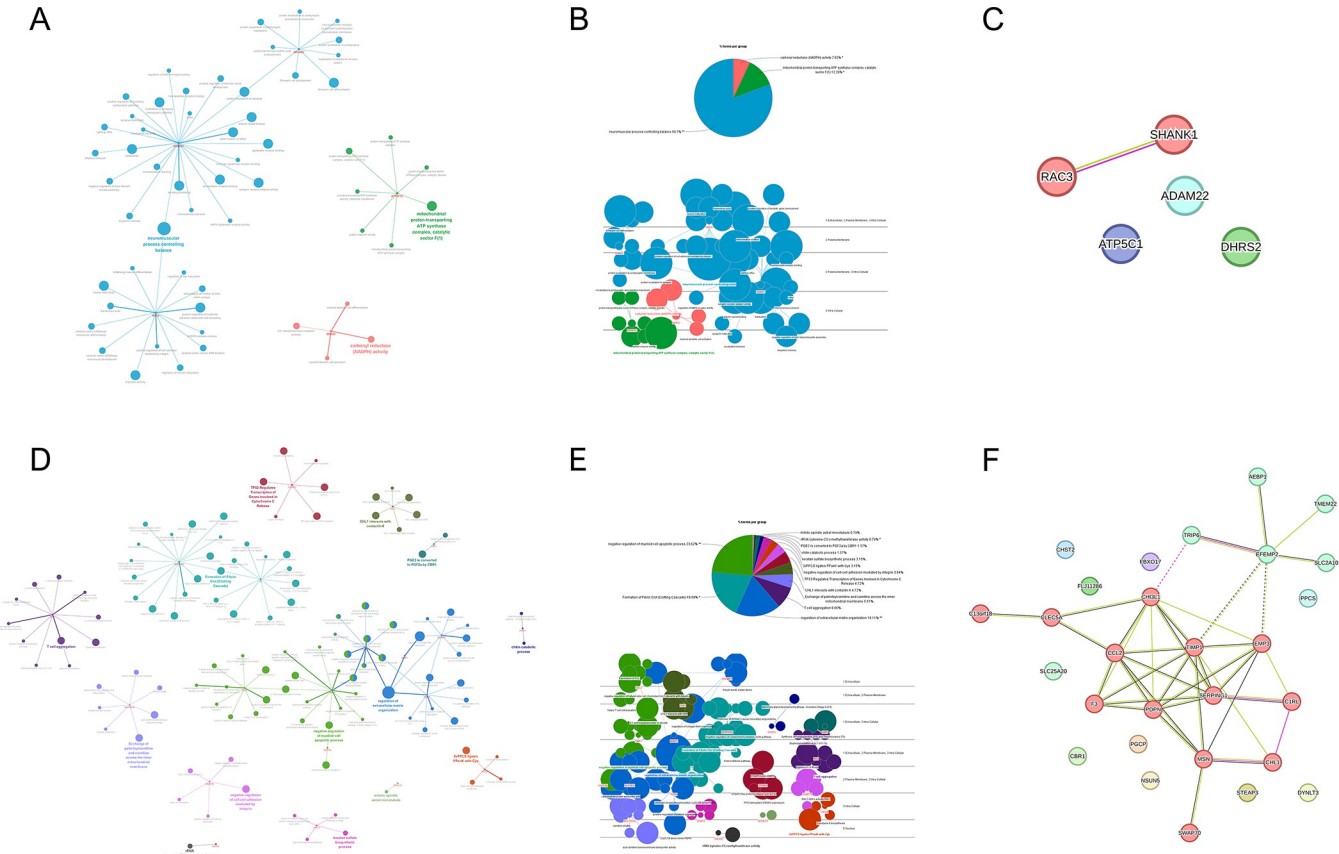

**Fig 4. Bioinformatic analysis of the significant genes associated with the prognosis of GBM using Cytoscape with ClueGo and CluePedia plugins and STRING database.** (A) Grouping of the networks of the significant genes associated with an improved prognosis of GBM based on functionally enriched GO terms and pathways using the ClueGo and CluePedia plugins of Cytoscape; (B) functionally grouped networks based on the GO terms of the genes significantly associated with an improved prognosis of GBM, showing three significant GO terms. The cerebral view shows the approximate location of those significant proteins in the cell; (C) a protein-protein interaction network was constructed among the genes associated with an improved prognosis of GBM; (D) grouping of the networks of the genes significantly associated with a poorer prognosis of GBM, based on functionally enriched GO terms and pathways using the ClueGo and CluePedia plugins of Cytoscape; (E) functionally grouped networks based on the GO terms of the genes significantly associated with a poorer prognosis of GBM, showing 14 significant GO terms. The cerebral view shows the approximate location of the significant proteins in the cell; (F) a protein-protein interaction network was constructed among the genes associated with a poorer prognosis of GBM, showing that they were roughly divided into two clusters.

GBM: glioblastoma multiforme; GO: gene ontology; STRING: Search Tool for the Retrieval of Interacting Genes/Proteins.

When protein-protein interaction was analyzed using STRING, only RAC3 and SHANK1 demonstrated a significant interaction (**Fig 4C**). There were 14 significant GO terms for the genes associated with poor prognosis in patients with GBM (**Fig 4D and 4E**). Among the 14 GO terms, the top four significant GO terms were 'negative regulation of myeloid cell apoptotic process', 'formation of fibrin clot (clotting cascade)', 'regulation of extracellular matrix organization', and 'T-cell aggregation' (**Fig 4D and 4E**). Following further analysis of the protein-protein interactions between the 28 genes associated with poor prognosis in patients with GBM, we found that the genes were roughly divided into two clusters (**Fig 4F**). These findings and possible mechanisms for the 33 significant genes affecting the OS and PFS in patients with GBM based on previous studies are summarized (**Table 2**).

## Discussion

In this study, we identified 33 independent genes, among 12,042 genes from the TCGA database, whose expressions were significantly associated with the prognosis of 525 patients with GBM. Among them, the expressions of five genes were independently associated with an improved prognosis of GBM, while the expressions of the other 28 genes were independently related to a worse prognosis of GBM. Moreover, the genes associated with long-term survival were identified in GBM patients. Among the five genes associated with an improved prognosis of GBM, the genes whose expressions were significantly associated with long-term survival of GBM patients were *ADAM22*, *ATP5C1*, *RAC3*, and *SHANK1*. Alternatively, among the 28 genes that were associated with a worse prognosis in GBM patients, the expressions of *AEBP1*, *C1RL*, *CHL1*, *CHST2*, *EFEMP2*, and *PGCP* were negatively related to the long-term survival of GBM patients. When bioinformatics analysis was performed, there were three significant GO terms among the genes associated with an improved prognosis of GBM, whereas, 14 significant GO terms were among genes associated with a worse prognosis of GBM.

We classified the 33 significant genes according to their GO terms and the possible roles of those proteins on the prognosis of GBM based on the GeneCards database (www.genecards. org) and previous studies (**Table 2**) [10–44]. GeneCards is known as a comprehensive, authoritative compendium of annotative information about human genes, which are automatically mined and integrated from over 80 digital sources, resulting in a web-based deep-linked card for each of $> 73\,000$ human gene entries [45].

Consequently, we found that the expression of the genes involved in the GBM immune microenvironment most commonly influences the GBM prognosis. To support this, our study showed significant correlations between the expressions of all 32 significant genes (except ATP5C1) and CD8+ T-cell infiltrations in the 525 GBM tissues. A recent study also reported that GBM cases with high-risk scores were involved in immune and inflammatory processes or pathways [46]. Based on our investigation, among the 33 significant genes, there were 12 significant genes that appeared to be related to the GBM immune microenvironment and may affect the prognosis of GBM: *C1RL*, *CCL2*, *CHI3L1*, *CLEC5A*, *EMP3*, *FBXO17*, *MSN*, *SERPING1*, *STEAP3*, *SWAP70*, *TIMP1*, and *TMEM22*. According to our findings, these 12 genes were associated with a worse prognosis for GBM; therefore, we hypothesized that they might be involved in the immunosuppression of the GBM microenvironment. Our findings support this hypothesis since all of these 12 genes were negatively correlated with CD8+ T-cell infiltrations in the GBM tissues. Moreover, we observed that these 12 genes are almost identical to the genes belonging to the red cluster in **Fig 4F**. The immune microenvironment of GBM is highly immunosuppressive due to the lack of a number of tumor-infiltrating lymphocytes and other

**Table 2. Classification of the 33 significant genes according to their GO terms alongside the possible mechanisms of the 33 significant proteins affecting the OS and PFS in GBM patients.**

| Gene symbol | Associated GO terms according to the Cytoscape analysis | Summary of the possible mechanisms of the 33 significant proteins affecting the OS and PFS in GBM patients | Classifications according to possible functional roles of the proteins in GBM | References |
|---|---|---|---|---|
| **1. Genes associated with enhanced OS and PFS** | | | | |
| ADAM22 | Neuromuscular process controlling balance | ADAM22, a brain-specific cell surface protein, mediates glioma growth inhibition using an integrin-dependent pathway. | Cell adhesion | [8] |
| RAC3 | | Although it is known as an oncogene, it has been reported that it plays the opposite role in glioma. RAC3 interacts with the integrin-binding protein and promotes integrin-mediated adhesion and spreading. Some integrins can promote the entry of adenoviral complexes into glioma stem cells and produce killing effects. Although the exact mechanism is unclear, we speculate that RAC3 may have tumor suppressive effects in an integrin-dependent manner in glioblastoma. | Cell adhesion, structural and extracellular matrix | [11–13] |
| SHANK1 | | SHANK1 acts as a negative regulator of integrin activity and consequently interferes with cell adhesion, spreading, migration, and invasion. | Cell adhesion, structural and extracellular matrix | [14] |
| ATP5C1 | Mitochondrial proton-transporting ATP synthase complex, catalytic sector F(1) | A common event in tumor cells is the metabolic switch from respiration (in the mitochondria) to glycolysis (in the cytosol), often referred to as "the Warburg effect". The increased expression of ATP5C1 may be associated with maintaining the activities of ATP synthase and cellular respiration leading to the inhibition of tumor progression. | Mitochondrial ATP synthesis | [9] |
| DHRS2 | Carbonyl reductase (NADPH) activity | DHRS2 is known as a tumor-suppressor gene that belongs to the short-chain dehydrogenase/reductase family. DHRS2 decreases the NADP/NADPH ratio and induces ROS clearance in mitochondria. In addition, DHRS2 is reported to bind MDM2 and lead to the attenuation of MDM2-intermediated p53 degradation. | NADPH activity | [10] |
| **2. Genes associated with a worse OS and PFS** | | | | |
| AEBP1 | Regulation of extracellular matrix organization | The AEBP1 activates MAP kinase in adipocytes, leading to adipocyte proliferation and reducing adipocyte differentiation. AEBP1 may promote GBM cell proliferation, migration, and invasion by activating the classical NF-κB pathway, which stimulates the activity and expression of the MMP-9. | Structural and extracellular matrix | [15] |
| EFEMP2 | | EFEMP2 is a member of fibulins, which are a family of extracellular matrix glycoproteins. EFEMP2 may promote tumor invasion in glioma by regulating MMP-2 and MMP-9. | Structural and extracellular matrix | [24] |
| PDPN | | PDPN is associated with cell elongation, cell adhesion, migration, and tube formation by promoting the rearrangement of the actin cytoskeleton. PDPN may promote invasive capacity, migration, and the radio-resistance of GBM cells. | Cell adhesion, structural and extracellular matrix | [31] |
| SLC2A10 | Both regulation of extracellular matrix organization and negative regulation of myeloid cell apoptotic process | SLC2 genes encode glucose transporters. SLC2A10 is significantly highly expressed in GBM with a poor prognosis. | Transporter | [37] |
| CCL2 | Negative regulation of myeloid cell apoptotic process | CCL2 is a potential candidate chemokine to regulate the chemoattraction of Treg to glioma. CCL2 recruits Tregs and myeloid-derived suppressor cells as major contributors to the potently immunosuppressive glioma microenvironment. | Immune system process | [18] |
| CLEC5A | | CLEC5A is a myeloid specific gene and may promote immunosuppression, tumor angiogenesis and cancer cell invasion in GBM. | Immune system process | [22] |
| TIMP1 | | TIMP1 is a specific inhibitor of MMP. TIMP1 shows aberrant upregulation in different types of cancers. TIMP1 levels are positively related to increased immune infiltration levels of tumor-infiltrating lymphocytes and correlate with cancer progression in GBM. | Immune system process | [41] |

*(Continued)*

**Table 2.** (Continued)

| Gene symbol | Associated GO terms according to the Cytoscape analysis | Summary of the possible mechanisms of the 33 significant proteins affecting the OS and PFS in GBM patients | Classifications according to possible functional roles of the proteins in GBM | References |
|---|---|---|---|---|
| *F3* | Formation of fibrin clot (clotting cascade) | F3 encodes coagulation factor III, which is a cell surface glycoprotein promoting hypercoagulation status. The hypercoagulation status both increases the risk of thromboembolic events and influences the brain tumor biology, thereby promoting its growth and progression by stimulating intracellular signaling pathways. | Blood coagulation cascade | [26] |
| *SERPING1* | | SERPING1 encodes plasma protein involved in the regulation of the complement cascade, C1 inhibitor, and immune cell response. The C1 inhibitor can inactivate plasmin and tissue plasminogen activators to promote clot formation. SERPING1 might also drive the hypoxic phenotype of peri necrotic GBM leading to hypoxia-induced glioma stemness. | Blood coagulation cascade, immune system process | [35,36] |
| *CBR1* | PGE2 is converted to PGF2a by CBR1 | CBR1 inactivates highly reactive lipid aldehydes and may play a meaningful role in preserving cells from oxidative stress. Inhibition of CBR1 induces accumulation of intracellular ROS levels leading to an increase in mitotic catastrophe and mitotic arrest. Among patients treated with radiation, patients with low CBR1 expression showed an improved prognosis. CBR1 may be crucial for the survival of cancer cells after radiation and can be a good target for developing radiosensitizers. | NADPH activity | [17] |
| *CHI3L1* | Chitin catabolic process | CHI3L1 is associated with the inflammatory response and promotes the progression of GBM by secreting cytokines released from immune cells. CHI3L1 may contribute to the immunosuppressive microenvironment of GBM. Inhibition of CHI3L1 may reduce immunosuppression and overcome immunotherapy resistance in GBM. | Immune system process | [19] |
| *CHL1* | CHL1 interacts with contactin-6 | CHL1 is a member of the cell adhesion molecule L1 family and plays a fundamental role in the development and progression of cancers. CHL1 is associated with promoting the survival of glioma cells while inhibiting apoptosis of glioma cells via the PI3K/AKT signaling pathway. | Cell adhesion | [20] |
| *CHST2* | Keratan sulfate biosynthetic process | CHST family has been reported as an oncogene in various cancers. However, the role of CHST2 in GBM is largely unknown. CHST family significantly increases GBM cell proliferation through the WNT/β-catenin pathway. | Metabolism | [21] |
| *DYNLT3* | Mitotic spindle astral microtubule | DYNLT3 is a component of the cytoplasmic dynein complex and binds with the mitotic protein to control mitosis and meiosis progression. It was reported that the low expression of DYNLT3 was associated with longer survival in female patients. | Cell cycle | [23] |
| *MSN* | T-cell aggregation | MSN is a link between the actin cytoskeleton and the plasma membrane and controls T-cell differentiation via the TGF-β receptor. Upregulation of MSN expression in glioblastoma cells might be correlated with increases in cell proliferation, invasion, and migration through the Wnt/β-catenin pathway. | Immune system process | [28,29] |
| *NSUN5* | rRNA (cytosine-C5)-methyltransferase activity | NSUN5 is an enzyme with tumor-suppressor properties that undergoes epigenetic loss in gliomas leading to an overall depletion of protein synthesis. NSUN5 epigenetic inactivation is a hallmark of glioma patients with long-term survival. | Embryonic development | [30] |
| *PPCS* | 2xPPCS ligates PPanK with Cys | PPCS catalyzes the pathway in which phosphopantothenate reacts with ATP and cysteine to form phosphopantothenoylcysteine. Phosphopantothenoylcysteine is an intermediate in the biosynthetic pathway that converts pantothenate (vitamin B5). Vitamin B5 is the key precursor for the biosynthesis of coenzyme A (CoA) and CoA may act as an acyl group carrier to form acetyl-CoA. Acetyl-CoA promotes glioblastoma cell adhesion and migration through $Ca^{2+}$–NFAT signaling. | Metabolism | [33,34] |

*(Continued)*

**Table 2.** (Continued)

| Gene symbol | Associated GO terms according to the Cytoscape analysis | Summary of the possible mechanisms of the 33 significant proteins affecting the OS and PFS in GBM patients | Classifications according to possible functional roles of the proteins in GBM | References |
|---|---|---|---|---|
| SLC25A20 | Exchange of palmitoylcarnitine and carnitine across the inner mitochondrial membrane | SLC25A20 is a mitochondrial-membrane-carrier protein associated with the transport of acylcarnitines into the mitochondrial matrix for oxidation. The role of SLC25A20 in glioma is unclear. The human protein atlas shows that low expression of SLC25A20 is associated with longer survival of patients with glioma. | Transporter | [38] |
| STEAP3 | TP53 regulates the transcription of genes involved in cytochrome c release | STEAP3 is one of the ferroptosis-related genes, which are associated with immune-related factors and the p53 signaling pathway. STEAP3 promotes GBM growth and invasion and is associated with a poor prognosis in GBM patients. | Immune system process, apoptotic process, cell cycle | [39] |
| SWAP70 | Negative regulation of cell–cell adhesion mediated by integrin | SWAP-70 is a guanine nucleotide exchange factor that is involved in cytoskeletal rearrangement and regulation of migration and invasion of malignant tumors. SWAP-70 may promote GBM cell migration and invasion by regulating the expression of CD44s, which contributes to lymphocytes adhering to the extracellular matrix of the brain, penetrating the white matter, and continuing to spread. | Cell adhesion, structural and extracellular matrix, immune system process | [40] |
| C13orf18 | N/A | C13orf18 gene encodes a cysteine-rich protein that contains a putative zinc-RING and/or ribbon domain. The role of C13orf18 in GBM is unclear. | Autophagy | N/A |
| C1RL | | The C1RL protein cleaves prohaptoglobin in the endoplasmic reticulum. C1RL probably plays a crucial role in glioma immunosuppression. | Immune system process | [16] |
| EMP3 | | EMP3 is a tetraspanin membrane protein that represses the induction and function of cytotoxic T-lymphocytes. EMP3 is an important immunosuppressive factor for recruiting tumor-associated macrophages in GBM leading to suppression of T-cell infiltration to facilitate tumor progression. | Immune system process | [25] |
| FBXO17 | | FBXO17 is reported as an F-box protein associated with high-grade glioma. FBXO17 promotes cell proliferation, migration, and invasion of glioma development via the modulation of the AKT/GSK-3β/Snail signaling pathway. | Immune system process, metabolism | [27] |
| FLJ11286 | | FLJ11286, an interferon-stimulated gene, contains conserved cysteine residues and has homologues across the vertebrate taxon. The role of FLJ11286 in GBM is unclear. | Unknown | N/A |
| PGCP | | The PGCP (CPQ) gene encodes a metallopeptidase belonging to the M28 peptidase family. The human protein atlas shows that low expression of CPQ is associated with an enhanced survival rate in patients with glioma. | Structural and extracellular matrix | [32] |
| TMEM22 | | TREM22 encodes immune receptors. The role of TREM22 in GBM is unclear. | Immune system process | N/A |
| TRIP6 | | TRIP6 can regulate multiple signaling pathways including NF-κB, extracellular signal-regulated kinase, and PI3K/AKT. Increased levels of TRIP6 may promote tumorigenesis through the regulation of p27KIP1 and correlates with the poor survival of glioma patients. | Cell adhesion | [42] |

GBM: glioblastoma multiforme; OS: overall survival; PFS: progression-free survival; ATP: adenosine triphosphate; NADPH: nicotinamide adenine dinucleotide phosphate; ROS: reactive oxygen species; MDM2: mouse double minute 2 homolog; MAP kinase: mitogen-activated protein kinase; NF-κB: nuclear factor-κB; MMP: matrix metalloproteinase; PI3K: phosphoinositide 3-kinase; AKT: protein kinase B; TGF-β: transforming growth factor-β; NFAT: nuclear factor of activated T-cells; GSK-3β: glycogen synthase kinase-3β.

immune effector cells in the GBM microenvironment [21]. This immunosuppressive GBM microenvironment results in resistance to immunotherapy and promotes a poor prognosis in GBM patients. Among the 12 significant genes involved in the immunosuppression of GBM, CCL2 recruits Tregs and myeloid-derived suppressor cells, which play a critical role in the immunosuppressive glioma microenvironment [20]. High levels of CHI3L1 are positively related to the infiltration of Tregs, neutrophils, and resting NK cells, which induces limitations in the effective anti-tumor immune response to GBM [21]. In addition, EMP3 is an important immunosuppressive factor for recruiting tumor-associated macrophages in GBM, which induces suppression of T-cell infiltration and leads to tumor progression [27]. Furthermore, C1RL may play an immunosuppressive role in the pathogenesis of glioma by triggering the activation of haptoglobin and complement component 1 [18].

The second most common possible mechanism related to the effect these 33 significant genes could produce on the prognosis of GBM was through cell adhesion or structural and extracellular matrix. According to our findings, 10 genes including *ADAM22*, *AEBP1*, *CHL1*, *EFEMP2*, *PDPN*, *PGCP*, *RAC3*, *SHANK1*, *SWAP70*, and *TRIP6* appeared to influence the prognosis of GBM through mechanisms involving cell adhesion or structural and extracellular matrix. Among the genes associated with a good prognosis in GBM patients, ADAM22, RAC3, and SHANK1 are thought to inhibit GBM progression in an integrin-dependent manner [10, 13–16]. Meanwhile, based on our investigation, AEBP1, EFEMP2, and PGCP, which were negatively related to long-term survival in GBM patients are thought to affect the prognosis of GBM through matrix metalloproteinases (MMPs)-related mechanisms [17, 26, 34]. Low expressions of MMP9 in GBM tissues are associated with a good response to temozolomide and longer survival of patients with GBM [47]. In addition, CHL1, which is also negatively associated with long-term survival in GBM patients, promotes the survival of glioma cells by inhibiting the apoptosis of glioma cells via the phosphatidylinositol 3-kinase (PI3K)/AKT signaling pathway [22].

Meanwhile, among the 33 independent and significant genes, CHST2, PPCS, and FBXO17 were considered to influence the prognosis of GBM in relation to metabolism [23, 29, 35, 36]. The role of CHST2 in GBM is largely unknown, however, it is thought to have a negative influence on long-term survival in GBM patients in our study. Moreover, it has been previously reported that the CHST family may cause GBM cell proliferation through the WNT/β-catenin pathway [23]. Furthermore, according to our study, the genes related to the blood coagulation cascade, such as F3 and SERPING1, may affect the prognosis for GBM. F3 encodes coagulation factor III, which promotes hypercoagulation status. The hypercoagulation status increases the risk of thromboembolic events and promotes the growth and progression of brain tumors by stimulating intracellular signaling pathways [28]. In addition, according to our study, an increased expression of ATP5C1, which is involved in mitochondrial ATP synthesis, was significantly associated with the long-term survival of GBM patients. A metabolic switch from respiration (in the mitochondria) to glycolysis (in the cytosol) is a common feature in tumor cells. However, increased expression of ATP5C1 may also be related to maintaining the activities of ATP synthase and cellular respiration, which leads to the inhibition of tumor progression [11].

In summary, the overexpression of C1RL, CCL2, CHI3L1, CLEC5A, EMP3, FBXO17, MSN, SERPING1, STEAP3, SWAP70, TIMP1, and TMEM22 genes appears to influence the prognosis of patients with GBM by causing an immune-suppressive GBM microenvironment. Immunotherapy holds tremendous promise for revolutionizing cancer therapies, but the significant immunosuppression seen in patients with GBM inhibits the effectiveness of immunotherapy. Therefore, reversing this GBM-mediated immune suppression is critical to increase the effectiveness of immunotherapy for GBM [48]. Consequently, we believe it is meaningful

to validate whether blocking the above 12 genes, which are associated with immunosuppression in GBM, affects the prognosis of GBM in this study. Secondly, ADAM22, AEBP1, CHL1, EFEMP2, PDPN, PGCP, RAC3, SHANK1, SWAP70, and TRIP6 genes may impact the prognosis of GBM through mechanisms involving cell adhesion or structural and extracellular matrix. ADAM22, RAC3, and SHANK1 were associated with a favorable prognosis in patients with GBM, and the expression of the remaining genes was associated with a poor prognosis. Focal adhesion is at the center of signaling pathways crucial for tumor development and may mediate radioresistance, chemotherapy, and resistance to targeted therapy in glioma [49]. Consequently, we believe that the above cell adhesion-related genes associated with the GBM prognosis identified in this study may have clinical implications for the future treatment of GBM. Finally, our results demonstrate that CHST2, PPCS, and FBXO17 may influence the prognosis of GBM through metabolism pathways. CHST2 could impact the WNT/β-catenin pathway, F3 and SERPING1 through blood coagulation cascade, and ATP5C1 through mitochondrial ATP synthesis. Therefore, based on our findings, we are planning future in vitro and/or in vivo experiments to validate the relationship between the identified genes and GBM prognosis. We expect that future experimental studies may contribute to improving the treatment of GBM.

This study has several limitations: Firstly, we obtained all clinical and mRNA expression data from the TCGA database, which is retrospective. Thus, further planned studies are required to verify these results. However, since public TCGA data was used and all the raw data is presented as Supplementary Data 1, our results can be evaluated and validated by other researchers. Secondly, the fraction of immune cells in GBM was estimated using in silico flow cytometry-based analysis, although this may not accurately reflect the actual number of immune cells. Thirdly, the current findings were not verified through experimental analyses; therefore, further in vitro and/or in vivo studies are required. Fourth, there are missing clinical and mRNA expression data that were unavailable in the TCGA dataset, potentially influencing the results of the statistical analyses in the study. Lastly, this study is subject to potential bias because it only used data from a single TCGA database. Therefore, verifying the results in future studies using different databases is necessary.

## Conclusion

Overall, we investigated significant genes related to both length of OS and PFS in patients with GBM using a large-scale, open database. According to our findings, there were 33 independent genes among 12,042 human genes whose expressions were significantly associated with the prognosis of GBM. Among these 33 significant genes, the expressions of five genes were associated with an improved prognosis of GBM, while numerous other genes were related to a worse prognosis in patients with GBM. In addition, expressions of *ADAM22*, *ATP5C1*, *RAC3*, *SHANK1*, *AEBP1*, *C1RL*, *CHL1*, *CHST2*, *EFEMP2*, and *PGCP* genes were either positively or negatively related to the long-term survival of GBM patients. Although our findings are required to be validated in the future, we believe that they may contribute to improving the understanding of the mechanisms underlying the pathophysiology of GBM.

## Supporting information

**S1 Data. The clinical information and mRNA expression data from the TCGA database of 525 GBM cases.**
(XLSX)

**S1 Fig. Kaplan–Meier curves showing overall survival (OS) and progression-free survival (PFS) rates according to DHRS2, AEBP1, C1RL, CBR1, CCL2, CLEC5A, DYNLT3, EFEMP2, EMP3, F3, FBXO17, FLJ11286, MSN, NSUN5, PDPN, and PGCP expression quartiles.**
(TIF)

**S2 Fig. Kaplan–Meier curves showing overall survival (OS) and progression-free survival (PFS) rates according to PPCS, SERPING1, SLC25A20, SLC2A10, STEAP3, SWAP70, TIMP1, TMEM22, and TRIP6 expression quartiles.**
(TIF)

**S1 Table. Clinical and immune cell characteristics in patients with GBM.**
(DOCX)

**S2 Table. Univariable linear regression analysis of the lengths of the OS and PFS according to the 36 significant gene expressions in patients with GBM.**
(DOCX)

## Author Contributions

**Conceptualization:** Myung-Hoon Han.

**Data curation:** Hong Gyu Yoon, Myung-Hoon Han.

**Formal analysis:** Hong Gyu Yoon, Myung-Hoon Han.

**Funding acquisition:** Myung-Hoon Han.

**Investigation:** Myung-Hoon Han.

**Methodology:** Myung-Hoon Han.

**Resources:** Jin Hwan Cheong.

**Software:** Hong Gyu Yoon.

**Supervision:** Jin Hwan Cheong, Je Il Ryu, Yu Deok Won, Kyueng-Whan Min.

**Validation:** Myung-Hoon Han.

**Visualization:** Myung-Hoon Han.

**Writing – original draft:** Hong Gyu Yoon, Myung-Hoon Han.

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
