## [Decision Letter · Decision Letter 0]

8 Nov 2023

PONE-D-23-30860The genes significantly associated with an improved prognosis and long-term survival of glioblastomaPLOS ONE

Dear Dr. Han,

Thank you for submitting your manuscript to PLOS ONE. After careful consideration, we feel that it has merit but does not fully meet PLOS ONE’s publication criteria as it currently stands. Therefore, we invite you to submit a revised version of the manuscript that addresses the points raised during the review process.

We look forward to receiving your revised manuscript.

Kind regards,

Syed M. Faisal, Ph.D.

Academic Editor

PLOS ONE

Journal Requirements:

 "This study was funded by the Basic Science Research Program through the National Research Foundation of Korea (NRF) funded by the Ministry of Science, ICT & Future Planning (NRF-2022R1F1A1063739)." 

"This study was funded by the Basic Science Research Program 

through the National Research Foundation of Korea (NRF) funded by the Ministry of Science, 

ICT & Future Planning (NRF-2022R1F1A1063739)."

 "This study was funded by the Basic Science Research Program through the National Research Foundation of Korea (NRF) funded by the Ministry of Science, ICT & Future Planning (NRF-2022R1F1A1063739)."

**Additional Editor Comments:**

Your study on identifying genes associated with the prognosis of patients with Glioblastoma Multiforme (GBM) using TCGA data is a valuable contribution to the field. However, to ensure the highest quality and impact of your work, we kindly ask you to address the following comments and suggestions from the reviewers:

 Please elaborate on the preprocessing steps of the TCGA gene expression data, including details on normalization and quality control, as well as the methods used to determine the significance threshold and adjust for multiple comparisons. Additionally, clarify how confounding factors were incorporated into the survival analysis. In terms of clinical relevance, an expanded discussion on how the identified genes might influence patient care would significantly strengthen your paper. Also, the structural flow, particularly in the Introduction, Methodology, and Results sections, requires improvement for better readability. Please address potential biases from using a single database and the statistical approach to managing multiple testing.  

Reviewers' comments:

Reviewer's Responses to Questions

**Comments to the Author**

1. Is the manuscript technically sound, and do the data support the conclusions?

Reviewer #1: Yes

Reviewer #2: Yes

2. Has the statistical analysis been performed appropriately and rigorously? 

Reviewer #1: Yes

Reviewer #2: Yes

3. Have the authors made all data underlying the findings in their manuscript fully available?

Reviewer #1: Yes

Reviewer #2: Yes

4. Is the manuscript presented in an intelligible fashion and written in standard English?

Reviewer #1: Yes

Reviewer #2: Yes

5. Review Comments to the Author

Reviewer #1: The study aimed to identify genes associated with the prognosis of patients with Glioblastoma Multiforme (GBM) using a large-scale, open database (The Cancer Genome Atlas - TCGA). The authors analyzed gene expression data and clinical information from 525 GBM patients and investigated the relationship between gene expression and both overall survival (OS) and progression-free survival (PFS) in GBM.

Methodology and Data Analysis:

Can you provide more details about the preprocessing steps of the gene expression data from the TCGA database, such as data normalization and quality control procedures?

How did you determine the significance threshold for identifying genes associated with prognosis? Did you correct for multiple testing, and if so, which method was used?

In the survival analysis, did you consider potential confounding factors such as age, gender, or treatment modalities? How were these factors addressed in your analysis?

Clinical Relevance:

What are the potential clinical implications of the identified genes associated with GBM prognosis? How might these findings impact patient care or treatment strategies?

Considering the retrospective nature of the study, what steps are planned for the validation of the identified genes and their associations with GBM prognosis in prospective studies or experimental settings?

Discussion and Interpretation:

The discussion section connects genes to potential mechanisms but could benefit from more context on how these findings fit into the broader landscape of GBM research. How do these results align with or contribute to existing knowledge in the field?

Structural Improvements:

Do you have any plans to enhance the structural clarity of the article, particularly in the Introduction, Methodology, and Results sections, to improve the overall flow and readability?

Statistical Considerations:

How did you address potential biases that could arise from using data from a single database, and are there any limitations associated with this approach?

Given that multiple statistical tests were performed, can you provide more information about how you corrected for multiple testing to control the family-wise error rate or false discovery rate?

Reviewer #2: The article entitled “The Genes Significantly Associated with an Improved Prognosis and Long-Term Survival of Glioblastoma” is an informative piece of work presented by the authors. As the title suggests, authors have identified some of the genes from the available data whose expressions are significantly related to the overall survival (OS) and progression-free survival (PFS) in glioblastoma (GBM) patients.

Elaborating on this retrospective study, the data for the mRNA expression of around 12K genes were retrieved from the TCGA database of 525 GBM tissues. Independent genes significantly associated with the prognosis of GBM were identified. ROC curve analysis was employed for the prediction of significant genes associated with the long-term survival of GBM patients. The authors identified 33 genes, among which the expression of 5 genes was independently associated with improved prognosis and the remaining 28 with poor prognosis. Some of the genes were either positively or negatively associated with the long-term survival of GBM patients.

This is a simple but informative study. A schematic flow chart depicting the overall process of identifying the gene is presented in Figure 1. The lengths of the OS and PFS according to the significant genes are given in a tabular form. The 33 selected genes were further considered for bioinformatic analysis.

Authors have employed appropriate statistical Analysis and bioinformatic tools for the analysis of different types of lymphocytes infiltrating the GBM tissues, correlations between the expressions of the 33 genes, the lengths of the OS and PFS, the immune cells in GBM, associations between the expressions of the selected genes and the lengths of the OS and PFS in GBM patients, and functional gene ontology and pathway network analyses.

Classification of the 33 significant genes according to their GO terms alongside the possible mechanisms of the 33 significant proteins affecting the OS and PFS in GBM patients in the form of the table is a good effort by the authors to put forward the findings in a simple and easy-to-understand manner.

The major limitation of the study is that it is completely computational, and nothing has been verified or validated by any supporting wet lab experiments. The article would have carried more weight if a couple of GBM cell lines had been used to simply look for the expression of the identified genes and observe whether they corroborate with the in-silico findings. But, as this is a nice attempt in this direction, and overall, the manuscript is well-planned, nicely executed, and written clearly and explicitly, I would recommend acceptance in its current form.

6. PLOS authors have the option to publish the peer review history of their article (what does this mean?). If published, this will include your full peer review and any attached files.

Reviewer #1: No

Reviewer #2: No

---

## [Author Response · Author response to Decision Letter 0]

13 Nov 2023

The genes significantly associated with an improved prognosis and long-term survival of glioblastoma

Hong Gyu Yoon, Jin Hwan Cheong, Je Il Ryu, Yu Deok Won, Kyueng-Whan Min, Myung-Hoon Han

Responses to reviewer comments

Editorial Board

PLOS ONE

Dear reviewers and editorial staff of PLOS ONE:

We are sincerely thankful for your thorough review of our manuscript. Based on the reviewers’ constructive suggestions, the critical issues from their analysis of our manuscript have been understood and addressed. 

These responses and revisions result from the authors’ hard work and sincerity in addressing the reviewers’ and editors’ suggestions. We are grateful for your guidance in improving our work’s scientific and literary quality.

This manuscript has not been published or presented elsewhere in part or entirety and is not under consideration by another journal. Given the study’s retrospective nature, the need for informed consent was waived, and participants’ data were anonymized. The appropriate ethics review board approved the study design. We have read and understood your journal’s policies and believe that neither the manuscript nor the study violates these. There are no conflicts of interest to declare.

Thank you for being so considerate. I look forward to hearing from you.

Sincerely,

Myung-Hoon Han, M.D., Ph.D.

Department of Neurosurgery, Hanyang University Guri Hospital, 153 Gyeongchun-ro, 

Guri, Gyonggi-do, Republic of Korea

Tel: +82-31-560-2326

Fax: +82.31-560-2327

E-mail: gksmh80@gmail.com

Reviewer #1

Methodology and Data Analysis: 

Can you provide more details about the preprocessing steps of the gene expression data from the TCGA database, such as data normalization and quality control procedures?

Thank you for your valuable comment here. We have added relevant sentences in the Methods, per Reviewer #2’s suggestions.

Methods

Study patients 

We obtained 1,149 glioma cases, consisting of 619 GBM cases and 530 low-grade glioma cases with mRNA gene expression data from the TCGA database (https://gdc.cancer.gov/about-data/publications/pancanatlas and https://www.cbioportal.org/) [5]. We initially selected 594 GBM cases with virtual histopathological slides and clinical data out of 619 GBM cases. We excluded 594 GBM cases with significantly incomplete mRNA gene expression information and clinical data. Therefore, the 525 GBM cases with complete virtual histopathological slides, mRNA expression data, and clinical information were finally included in the study as described elsewhere [3,4]. Log 2 (x + 1) transformation normalized all mRNA gene expression values before analysis [6]. As previously described [3,4], the identical 525 GBM cases with the virtual histopathological slides, clinical information, and mRNA gene expression data obtained from TCGA were used to perform this study (https://www.cbioportal.org/). 

Informed consent was not required because the data were obtained from the publicly available TCGA database.

How did you determine the significance threshold for identifying genes associated with prognosis? Did you correct for multiple testing, and if so, which method was used?

Thank you, we understand your comment. Using the methods described in the Study Design section of Methods and illustrated in Figure 1, we determined a significance threshold for identifying genes associated with GBM prognosis. 

When we compare treatment groups multiple times, the probability of finding a difference just by chance increases depending on the number of times we compare [1]. However, our study did not use statistics comparing two or more groups. This study did not perform multiple statistical tests to compare the expression levels of various genes between the experimental and control groups. However, it simply analyzed the correlation between the length of OS or PFS and the expression levels of genes. Therefore, we believe that the statistical analysis in this study is not related to errors caused by multiple testing [1].

First, we excluded 11,187 genes from 12,042 genes whose expressions showed no significant association with the lengths of the OS or PFS in patients with GBM, according to Pearson correlation analysis (p ≥ 0.01).

Second, according to a previous study [2], we excluded 819 genes with a low correlation strength: Genes showing a Pearson coefficient absolute value of less than 0.2.

Third, after adjusting for clinical variables such as sex, age, Karnofsky performance scale score, and radiation treatment in the multivariable linear regression analysis, we finally identified 33 independent genes. These genes exhibited a significant association between their mRNA expressions and the length of both OS and PFS in patients with GBM, as shown in Table 1.

References

1. Ranganathan P, Pramesh CS, Buyse M. Common pitfalls in statistical analysis: The perils of multiple testing. Perspect Clin Res. 2016;7: 106–107. doi:10.4103/2229-3485.179436

2. Zou KH, Tuncali K, Silverman SG. Correlation and Simple Linear Regression. Radiology. 2003;227: 617–628. doi:10.1148/radiol.2273011499

In the survival analysis, did you consider potential confounding factors such as age, gender, or treatment modalities? How were these factors addressed in your analysis?

Thank you for your comment. As addressed in the preceding response, the 33 genes associated with prognosis in GBM are independent genes that were identified through multivariable linear regression analysis adjusted for sex, age, Karnofsky performance scale score, and radiation treatment, as shown in Table 1 (when adjusting for sex, age, Karnofsky performance scale score, and radiation treatment, three of the 36 genes lost statistical significance and were excluded). We have added the results of the univariate linear regression analysis of the lengths of the OS and PFS according to the 36 significant gene expressions in patients with GBM to the revised manuscript as S2 Table, as shown below.

S2 Table. Univariable linear regression analysis of the lengths of the OS and PFS according to the 36 significant gene expressions in patients with GBM

 Univariable linear regression analysis

 Length of OS (months) Length of PFS (months)

Variable β (95% CI) p-value β (95% CI) p-value

ADAM22 16.94 (10.33 to 23.55) < 0.001 13.78 (9.02 to 18.54) < 0.001

AEBP1 –2.81 (–3.83 to –1.78) <0.001 –1.93 (–2.67 to –1.18) <0.001

ATP5C1 9.21 (6.08 to 12.33) <0.001 5.59 (3.30 to 7.87) <0.001

C13orf18 –3.38 (–4.71 to –2.05) <0.001 –2.74 (–3.69 to –1.78) <0.001

C1RL –3.64 (–4.98 to –2.30) <0.001 –2.41 (–3.38 to –1.43) <0.001

CBR1 –3.74 (–5.18 to –2.30) <0.001 –3.19 (–4.22 to –2.16) <0.001

CCL2 –1.93 (–2.73 to –1.13) < 0.001 –1.39 (–1.97 to –0.81) < 0.001

CHI3L1 –1.58 (–2.21 to –0.95) <0.001 –1.47 (–1.92 to –1.02) <0.001

CHL1 –2.39 (–3.23 to –1.55) <0.001 –1.94 (–2.54 to –1.33) <0.001

CHST2 –3.49 (–4.86 to –2.11) <0.001 –2.57 (–3.56 to –1.57) <0.001

CLEC5A –3.01 (–4.17 to –1.84) <0.001 –2.34 (–3.18 to –1.49) <0.001

DHRS2 8.20 (5.59 to 10.80) <0.001 4.79 (2.88 to 6.71) <0.001

DYNLT3 –4.50 (–6.07 to –2.94) < 0.001 –3.98 (–5.10 to –2.87) < 0.001

EFEMP2 –5.02 (–6.37 to –3.68) <0.001 –3.39 (–4.37 to –2.40) <0.001

EMP3 –2.60 (–3.63 to –1.57) <0.001 –2.25 (–2.99 to –1.51) <0.001

F3 –3.48 (–4.82 to –2.14) <0.001 –2.90 (–3.86 to –1.94) <0.001

FBXO17 –4.72 (–6.49 to –2.96) <0.001 –3.46 (–4.74 to –2.18) <0.001

FLJ11286 –4.25 (–5.91 to –2.59) <0.001 –3.23 (–4.43 to –2.03) <0.001

KIAA0495 –8.66 (–5.91 to –2.59) < 0.001 –6.74 (–8.97 to –4.51) < 0.001

MSN –4.73 (–6.31 to –3.14) <0.001 –3.42 (–4.57 to –2.27) <0.001

NSUN5 –4.73 (–6.31 to –3.14) <0.001 –4.31 (–5.86 to –2.77) <0.001

PDPN –2.32 (–3.23 to –1.42) <0.001 –1.89 (–2.54 to –1.24) <0.001

PGCP –5.09 (–6.95 to –3.22) <0.001 –3.83 (–5.18 to –2.48) <0.001

PPCS –5.15 (–7.23 to –3.06) <0.001 –4.26 (–5.76 to –2.75) <0.001

RAC3 8.47 (5.34 to 11.59) < 0.001 5.63 (3.36 to 7.90) < 0.001

RANBP17 12.42 (7.86 to 16.99) <0.001 8.85 (5.54 to 12.16) <0.001

RBP1 –2.15 (–3.04 to –1.26) <0.001 –1.93 (–2.57 to –1.29) <0.001

SERPING1 –2.80 (–3.85 to –1.75) <0.001 –1.86 (–2.63 to –1.09) <0.001

SHANK1 18.47 (11.34 to 25.60) <0.001 13.38 (8.21 to 18.54) <0.001

SLC25A20 –5.40 (–7.21 to –3.59) <0.001 –3.60 (–4.92 to –2.28) <0.001

SLC2A10 –3.25 (–4.52 to –1.98) <0.001 –2.61 (–3.52 to –1.69) <0.001

STEAP3 –3.17 (–4.47 to –1.86) <0.001 –2.88 (–3.82 to –1.95) <0.001

SWAP70 –4.41 (–6.21 to –2.61) <0.001 –3.23 (–4.53 to –1.93) <0.001

TIMP1 –3.13 (–4.35 to –1.90) <0.001 –2.82 (–3.69 to –1.94) <0.001

TMEM22 –3.57 (–4.91 to –2.23) <0.001 –2.51 (–3.49 to –1.54) <0.001

TRIP6 –3.72 (–5.25 to –2.20) <0.001 –2.70 (–3.80 to –1.59) <0.001

OS: overall survival; PFS: progression-free survival; CI: confidence interval

We also added relevant sentences in the Study Design section of Methods as follows.

Methods

Study design

(5) A total of 33 genes whose expressions showed significant independent associations with both the lengths of the OS and PFS in patients with GBM were finally enrolled for the study. We also present the results of the univariate linear regression analysis of the lengths of the OS and PFS according to the 36 significant gene expressions in patients with GBM in the S2 Table. The raw data related to the study design can be found in the S1 Data. 

Clinical Relevance:

What are the potential clinical implications of the identified genes associated with GBM prognosis? How might these findings impact patient care or treatment strategies?

Considering the retrospective nature of the study, what steps are planned for the validation of the identified genes and their associations with GBM prognosis in prospective studies or experimental settings?

Discussion and Interpretation:

The discussion section connects genes to potential mechanisms but could benefit from more context on how these findings fit into the broader landscape of GBM research. How do these results align with or contribute to existing knowledge in the field?

Thank you, and we acknowledge your comment. We have previously summarized the possible mechanisms and functional roles related to the prognosis of GBM of the 33 identified genes in Table 2 and the Discussion section. The discussion section of the revised manuscript has been updated with new summarizing sentences related to the potential clinical implications of our results for the prognosis of GBM.

In addition, as commented by Reviewer #1, we are planning future in vitro, in vivo, or both experiments to validate the relationship between the identified genes and GBM prognosis based on our findings. We expect that future experimental studies may contribute to improving the treatment of GBM.

Furthermore, as suggested by Reviewer #1, we have added sentences related to applying our results to existing knowledge of GBM research.

Therefore, relevant sentences covering all the above concerns have been added as follows to the discussion section of the revised manuscript:

Discussion

However, increased expression of ATP5C1 may also be related to maintaining the activities of ATP synthase and cellular respiration, which leads to the inhibition of tumor progression [11].

In summary, the overexpression of C1RL, CCL2, CHI3L1, CLEC5A, EMP3, FBXO17, MSN, SERPING1, STEAP3, SWAP70, TIMP1, and TMEM22 genes appears to influence the prognosis of patients with GBM by causing an immune-suppressive GBM microenvironment. Immunotherapy holds tremendous promise for revolutionizing cancer therapies, but the significant immunosuppression seen in patients with GBM inhibits the effectiveness of immunotherapy. Therefore, reversing this GBM-mediated immune suppression is critical to increase the effectiveness of immunotherapy for GBM [48]. Consequently, we believe it is meaningful to validate whether blocking the above 12 genes, which are associated with immunosuppression in GBM, affects the prognosis of GBM in this study. Secondly, ADAM22, AEBP1, CHL1, EFEMP2, PDPN, PGCP, RAC3, SHANK1, SWAP70, and TRIP6 genes may impact the prognosis of GBM through mechanisms involving cell adhesion or structural and extracellular matrix. ADAM22, RAC3, and SHANK1 were associated with a favorable prognosis in patients with GBM, and the expression of the remaining genes was associated with a poor prognosis. Focal adhesion is at the center of signaling pathways crucial for tumor development and may mediate radioresistance, chemotherapy, and resistance to targeted therapy in glioma [49]. Consequently, we believe that the above cell adhesion-related genes associated with the GBM prognosis identified in this study may have clinical implications for the future treatment of GBM. Finally, our results demonstrate that CHST2, PPCS, and FBXO17 may influence the prognosis of GBM through metabolism pathways. CHST2 could impact the WNT/β-catenin pathway, F3 and SERPING1 through blood coagulation cascade, and ATP5C1 through mitochondrial ATP synthesis. Therefore, based on our findings, we are planning future in vitro and/or in vivo experiments to validate the relationship between the identified genes and GBM prognosis. We expect that future experimental studies may contribute to improving the treatment of GBM.

This study has several limitations: Firstly, we obtained all clinical and mRNA expression data from the TCGA database, which is retrospective. Thus, further planned studies are required to verify these results.

Structural Improvements:

Do you have any plans to enhance the structural clarity of the article, particularly in the Introduction, Methodology, and Results sections, to improve the overall flow and readability?

We have tried to improve the revised manuscript’s structural clarity as suggested by Reviewer #1.

Statistical Considerations:

How did you address potential biases that could arise from using data from a single database, and are there any limitations associated with this approach? 

Thank you, and we appreciate your comment. We acknowledge the potential biases mentioned by Reviewer #1, particularly regarding the use of data from a single TCGA database. We will conduct a study in the future to validate this study’s results using multiple databases. In the revised manuscript, we have added a sentence to the limitation section, acknowledging as follows that there may be a bias because we only used a single TCGA database:

Discussion

Fourth, there are missing clinical and mRNA expression data that were unavailable in the TCGA dataset, potentially influencing the results of the statistical analyses in the study. Lastly, this study is subject to potential bias because it only used data from a single TCGA database. Therefore, verifying the results in future studies using different databases is necessary.

Given that multiple statistical tests were performed, can you provide more information about how you corrected for multiple testing to control the family-wise error rate or false discovery rate?

Type 1 error is the chance of finding a difference between them just by chance when comparing two or more groups. When we compare treatment groups multiple times, the probability of finding a difference just by chance increases depending on the number of times we compare [1]. However, our study did not use statistics comparing two or more groups. This study did not perform multiple statistical tests to compare the expression levels of various genes between the experimental and control groups. However, it simply analyzed the correlation between the length of OS or PFS and the expression levels of genes. Therefore, we believe that the statistical analysis in this study is not related to errors caused by multiple testing [1].

Reference

1. Ranganathan P, Pramesh CS, Buyse M. Common pitfalls in statistical analysis: The perils of multiple testing. Perspect Clin Res. 2016;7: 106–107. doi:10.4103/2229-3485.179436

Reviewer #2: The article entitled “The Genes Significantly Associated with an Improved Prognosis and Long-Term Survival of Glioblastoma” is an informative piece of work presented by the authors. As the title suggests, authors have identified some of the genes from the available data whose expressions are significantly related to the overall survival (OS) and progression-free survival (PFS) in glioblastoma (GBM) patients.

Elaborating on this retrospective study, the data for the mRNA expression of around 12K genes were retrieved from the TCGA database of 525 GBM tissues. Independent genes significantly associated with the prognosis of GBM were identified. ROC curve analysis was employed for the prediction of significant genes associated with the long-term survival of GBM patients. The authors identified 33 genes, among which the expression of 5 genes was independently associated with improved prognosis and the remaining 28 with poor prognosis. Some of the genes were either positively or negatively associated with the long-term survival of GBM patients.

This is a simple but informative study. A schematic flow chart depicting the overall process of identifying the gene is presented in Figure 1. The lengths of the OS and PFS according to the significant genes are given in a tabular form. The 33 selected genes were further considered for bioinformatic analysis.

Authors have employed appropriate statistical Analysis and bioinformatic tools for the analysis of different types of lymphocytes infiltrating the GBM tissues, correlations between the expressions of the 33 genes, the lengths of the OS and PFS, the immune cells in GBM, associations between the expressions of the selected genes and the lengths of the OS and PFS in GBM patients, and functional gene ontology and pathway network analyses.

Classification of the 33 significant genes according to their GO terms alongside the possible mechanisms of the 33 significant proteins affecting the OS and PFS in GBM patients in the form of the table is a good effort by the authors to put forward the findings in a simple and easy-to-understand manner.

The major limitation of the study is that it is completely computational, and nothing has been verified or validated by any supporting wet lab experiments. The article would have carried more weight if a couple of GBM cell lines had been used to simply look for the expression of the identified genes and observe whether they corroborate with the in-silico findings. But, as this is a nice attempt in this direction, and overall, the manuscript is well-planned, nicely executed, and written clearly and explicitly, I would recommend acceptance in its current form.

Thank you very much for your positive evaluation of our article. As written in the limitations of the paper, we also think that the main limitation of this study is that it is completely computational without lab experiments. As commented by Reviewer #2, we are planning future in vitro, in vivo, or both experiments to validate the relationship between the identified genes and GBM prognosis based on our findings. We expect that future experimental studies may contribute to improving the treatment of GBM. Accordingly, we have added relevant sentences to the discussion section of the revised manuscript.

Discussion

However, increased expression of ATP5C1 may also be related to maintaining the activities of ATP synthase and cellular respiration, which leads to the inhibition of tumor progression [11].

In summary, the overexpression of C1RL, CCL2, CHI3L1, CLEC5A, EMP3, FBXO17, MSN, SERPING1, STEAP3, SWAP70, TIMP1, and TMEM22 genes appears to influence the prognosis of patients with GBM by causing an immune-suppressive GBM microenvironment. Immunotherapy holds tremendous promise for revolutionizing cancer therapies, but the significant immunosuppression seen in patients with GBM inhibits the effectiveness of immunotherapy. Therefore, reversing this GBM-mediated immune suppression is critical to increase the effectiveness of immunotherapy for GBM [48]. Consequently, we believe it is meaningful to validate whether blocking the above 12 genes, which are associated with immunosuppression in GBM, affects the prognosis of GBM in this study. Secondly, ADAM22, AEBP1, CHL1, EFEMP2, PDPN, PGCP, RAC3, SHANK1, SWAP70, and TRIP6 genes may impact the prognosis of GBM through mechanisms involving cell adhesion or structural and extracellular matrix. ADAM22, RAC3, and SHANK1 were associated with a favorable prognosis in patients with GBM, and the expression of the remaining genes was associated with a poor prognosis. Focal adhesion is at the center of signaling pathways crucial for tumor development and may mediate radioresistance, chemotherapy, and resistance to targeted therapy in glioma [49]. Consequently, we believe that the above cell adhesion-related genes associated with the GBM prognosis identified in this study may have clinical implications for the future treatment of GBM. Finally, our results demonstrate that CHST2, PPCS, and FBXO17 may influence the prognosis of GBM through metabolism pathways. CHST2 could impact the WNT/β-catenin pathway, F3 and SERPING1 through blood coagulation cascade, and ATP5C1 through mitochondrial ATP synthesis. Therefore, based on our findings, we are planning future in vitro and/or in vivo experiments to validate the relationship between the identified genes and GBM prognosis. We expect that future experimental studies may contribute to improving the treatment of GBM.

This study has several limitations: Firstly, we obtained all clinical and mRNA expression data from the TCGA database, which is retrospective. Thus, further planned studies are required to verify these results.

---

## [Editor Report · Decision Letter 1]

15 Nov 2023

The genes significantly associated with an improved prognosis and long-term survival of glioblastoma

PONE-D-23-30860R1

Dear Dr. Han,

We’re pleased to inform you that your manuscript has been judged scientifically suitable for publication and will be formally accepted for publication once it meets all outstanding technical requirements.

Kind regards,

Syed M. Faisal, Ph.D.

Academic Editor

PLOS ONE
---

## [Editor Report · Acceptance letter]

17 Nov 2023

PONE-D-23-30860R1 

The genes significantly associated with an improved prognosis and long-term survival of glioblastoma 

Dear Dr. Han:

I'm pleased to inform you that your manuscript has been deemed suitable for publication in PLOS ONE. Congratulations! Your manuscript is now with our production department. 

Kind regards, 

on behalf of

Dr. Syed M. Faisal 

Academic Editor

PLOS ONE